# Pollution-Free and Highly Sensitive Lactate Detection in Cell Culture Based on a Microfluidic Chip

**DOI:** 10.3390/mi14040770

**Published:** 2023-03-30

**Authors:** Jiaming Shi, Wenqiang Tong, Zhihang Yu, Lei Tong, Huaying Chen, Jing Jin, Yonggang Zhu

**Affiliations:** 1School of Science, Harbin Institute of Technology (Shenzhen), Shenzhen 518000, China; 2Center for Microflows and Nanoflows, School of Mechanical Engineering and Automation, Harbin Institute of Technology (Shenzhen), Shenzhen 518000, China

**Keywords:** cell metabolism, microfluidics, fluid mechanics, high sensitivity, pollution-free

## Abstract

Cell metabolite detection is important for cell analysis. As a cellular metabolite, lactate and its detection play an important role in disease diagnosis, drug screening and clinical therapeutics. This paper reports a microfluidic chip integrated with a backflow prevention channel for cell culture and lactate detection. It can effectively realize the upstream and downstream separation of the culture chamber and the detection zone, and prevent the pollution of cells caused by the potential backflow of reagent and buffer solutions. Due to such a separation, it is possible to analyze the lactate concentration in the flow process without contamination of cells. With the information of residence time distribution of the microchannel networks and the detected time signal in the detection chamber, it is possible to calculate the lactate concentration as a function of time using the de-convolution method. We have further demonstrated the suitability of this detection method by measuring lactate production in human umbilical vein endothelial cells (HUVEC). The microfluidic chip presented here shows good stability in metabolite quick detection and can work continuously for more than a few days. It sheds new insights into pollution-free and high-sensitivity cell metabolism detection, showing broad application prospects in cell analysis, drug screening and disease diagnosis.

## 1. Introduction

Cellular metabolism involves complex sequences of controlled biochemical reactions in living organisms to grow, reproduce and respond to environmental changes. The chemical compounds involved in cellular metabolism have numerous functions, such as fuels, signaling, stimulatory and inhibitory effects on enzymes, structures and so on [1,2]. The analysis of cellular metabolites plays a critical role in disease diagnosis [3,4,5] and drug development [6,7,8]. There have been ever-increasing research efforts in this field [9,10,11,12,13,14]. Among these metabolites, lactate is the main product of anaerobic metabolism. It is an important human monitoring parameter in the intensive care unit, which can be used to predict the likelihood of shock, collapse and mortality [15,16], and it is also an important cancer marker [17,18]. To analyze the metabolites of cells, sample volumes need to be small to reflect the localized environment around the cell due to a cell’s small size (picoliters to nanoliters). However, the sample volume in most of the current cell analyses, e.g., those in vitro cell culturing using Petri dishes and well plates, is in the order of milliliters. It is not possible to capture the cellular microenvironment. While some high-resolution imaging techniques [19,20] can measure cellular components, these are not suitable for pollution-free applications.

Microfluidics are micron-scale fluidic components (e.g., pumps, valves, mixers, reactors) developed in the past three decades. Microfluidic chips are devices which usually integrate several of these components to perform certain tasks such as biochemical analysis and materials synthesis. One important application of microfluidics is the cultivation of cells within microfluidic chips, which presents a challenge to the traditional methods of cell culture based on Petri dish and culture flask. The microenvironment of cells can be controlled through the cell culture system based on microfluidic chips. Moreover, due to their easy integration, high degree of automation and powerful liquid handling capabilities, microfluidic chips have been widely used in cell metabolite analysis such as lactate detection [21,22,23,24,25,26,27]. However, the same as in the Petri dish system, the detection reagents used for metabolite detection could have adverse effects on target cells (e.g., culture solution pollution, inhibition of cell development and even death). This must be avoided, especially for cells to be re-used, such as embryo cells in IVF (in vitro fertilization).

Most microfluidic systems placed sensors around cells or even injected reagent solutions into cells directly [28,29,30,31]. For example, the in situ cell metabolism detection device proposed by Lin et al. placed a sensor on the cell culture channel [28]. The multiparameter microphysiological monitoring system designed by Weltin et al. put the detection electrode in the cell culture chamber [29]. A drop-based cell-generated-lactate detection chip developed by Mongersun et al. encapsulated the indicator and cells in the same droplet [30]. Although this is simple to operate and easy to detect, it inevitably affects cell growth. Thereby it will interfere with cell metabolism unavoidably, which is not conducive to long-term detection of cells. In addition, some devices culture cells in Petri dishes, and then part of the culture medium consumed by the cells is injected into the chip for detection to avoid cell pollution [32,33]. For instance, Urbanski et al. designed a multichannel non-invasive embryo metabolism analysis chip. It only has the function of metabolite detection, while embryo culture was then carried out in a culture dish [32]. Prill et al. developed a long-term microfluidic metabolic monitoring chip. The target cells were cultured in a culture dish, and then the culture medium was transferred to a microfluidic chip for detection [33]. While this detection method avoided the pollution of cells by the detection reagents, it required manual intervention. This method not only significantly limits the application of this technique, but also limits the integration of other complimentary techniques, such as photographic observation of cells.

In order to achieve pollution-free and highly sensitive cell metabolite detection, it is necessary to separate cell culturing and metabolite detection on the same microfluidic chip. With this purpose, this paper proposes a microfluidic chip for measuring the real-time lactate production of cells. This chip integrates cell culturing and detection chambers with a backflow prevention channel to avoid the pollution of cell culturing from the detection zone. The concentration of lactate and its variation with time are determined from the fluorescence detection signal and the residence time distributions of the microchannel networks using the developed signal de-convolution algorithm. Both numerical simulation and experimental methods are used to determine the residence time distributions.

This study demonstrates that the proposed chip can realize pollution-free and highly sensitive detection of lactate produced by cells, exhibiting great application potential for cell analysis, such as drug screening, disease diagnosis and environmental monitoring.

## 2. Materials and Methods

### 2.1. Lactate Measurements

The microfluidic cell culture and detection chip is schematically shown in Figure 1. In operation, the medium in the culture chamber is first pushed out, mixed with the indicator in the microchannel to react, and then injected into the downstream detection zone for fluorescence detection. Afterwards, a fresh buffer solution is injected into the chip to flush the system and prevent the residual liquid from interfering with the next round of experiments. 

In this study, the lactate is measured by a laser-induced fluorescence method. It is described as follows. The lactate is mixed with the reagent solution (Figure 1), i.e., lactate oxidase (LOX, Yuanye Bio-Technology, Shanghai, China) and catalyzed to produce hydrogen peroxide (H_2_O_2_). The latter reacts with Amplex Red (10-acetyl-3,7-dihydroxyphenoxazine, Thermo Fisher, Wilmington, MA, USA) to produce red fluorescent Resorufin in the presence of horseradish peroxidase (HRP, Sigma Aldrich, St. Louis, MO, USA), added as a catalyst together with lactate oxidase. The generated fluorescence is collected by the objective lens on a microscope and then transmitted to a photoelectric detector, where it is converted into an electrical signal and then transmitted and recorded on a computer. The maximum excitation wavelength and emission wavelength of Resorufin are 571 and 585 nm, respectively.

### 2.2. Device Design and Fabrication

The microfluidic chip shown in Figure 1 consists of a PDMS layer with the micro-channel structure, which was fabricated by soft lithography technology. The manufacturing process of the microfluidic chip involved several steps. Firstly, SU-8 2075 (Microchem, Newton, MA, USA) was spin-coated on a thin silicon wafer, and then the male mold was made using reverse diffusion lithography technology. Next, the casting pre-polymer (Sylgard 184, Dow Corning, Midland, MI, USA) was prepared by mixing a substrate and a curing agent with a mass ratio of 1:10, which was then cast on the positive film to prepare a PDMS plate. The prepolymer was cured at 60 °C for 4 h and then punched with a hole punch. Afterwards, the PDMS plate and the glass plate substrate were treated with oxygen plasma for 30 s using a plasma cleaner (PDC-002, Harrick Plasma, Ithaca, NY, USA) to form covalent bonds. Finally, the two materials were quickly combined and heated at 60 °C for 4 h [34]. The microchip used in this paper had a microchannel size of 200 μm × 200 μm (height × width), a cell culture chamber that was about 1 mm wide and 3 mm long with a volume of 0.52 μL and a detection zone with a diameter of 1 mm (red circle in Figure 1).

### 2.3. Cell Seeding

The microfluidic chip was placed in a stage top incubator (Tokai Hit, Shizuoka-ken, Japan), which can control the internal environment of 5% carbon dioxide and 37 °C temperature for cell growth. In order to obtain a sterile surface inside the microfluidic chip, the syringe, the connecting tube and the microfluidic chip were rinsed with 75% ethanol for at least 20 min before cell seeding, and then rinsed with sterile deionized water for more than 10 min. In order to promote cell adhesion, the cell culture chamber was treated with fetal bovine serum (Every Green, Zhejiang, China) for 1 h. The trypsinized (Corning, Corning, NY, USA) cells (HUVEC, Otwo Biotech, Shenzhen, China), at a concentration of 10^6^ cells/mL, were then seeded on the microfluidic chip, and cell adhesion was observed under a microscope. The cells adhered to the cell culture chamber within 1 h. After cell adhesion, the cell solution was changed to cell culture medium. A glucose-free medium (Corning, Corning, NY, USA) was spiked with glucose (Sigma Aldrich, St. Louis, MO, USA) as the cell culture medium. The spiked glucose had a concentration of 500 μM. According to the results measured by Unterluggauer et al., the cellular glucose metabolism rate was 85 nmol/(hours × 10^5^ cells) [35]. Thus 500 μM glucose was sufficient for cell development.

### 2.4. Fluid Actuation

A syringe pump (CETONI, Bremen, Germany) was utilized to connect cell culture media, buffer (PBS, Thermo Fisher, Wilmington, MA, USA) and reagent solutions. Each detection cycle included the following three steps: (1) flushing: the syringe pump injected buffer solution into the microchip at a rate of 1 μL/min through the buffer inlet and flushed the detection zone for 20 min; (2) stopping: the syringe pump was turned off for 2 h. During the stopping step, the cells consume the supplied oxygen and glucose and then produce lactate; (3) detection: the syringe pumps of medium inlet and reagent solution inlet were opened simultaneously. The medium and reagent solution were injected into the microchip at a rate of 1 μL/min. The medium consumed by the cells in the culture chamber was quickly pushed to the downstream detection zone to react with the reagent solution, where the whole process lasted for around 20 min. At the end of a cycle, a picture of the cell culture chamber was taken using a camera (Photometrics Prime 95B, Tucson, AZ, USA), which was used to record survival status and record cell number. The physical environment of the chip (such as temperature, humidity, gas, etc.) was controlled by the stage top incubator on the microscope. The computer programming sequence automatically controlled each step of the syringe pump and the optical detection. It can avoid the error caused by manual operation and strictly controls the time of each step.

### 2.5. CFD Simulations

A 3D model was first created in Creo (PTC Inc., Boston, MA, USA) and then imported into COMSOL Multiphysics (COMSOL Inc., Burlington, MA, USA) for meshing and simulating (Figure 2), to simulate the convection-diffusion process of lactate on the microchip. Here we have conducted two simulations, one being the convection-diffusion simulation using the laminar flow and the dilute material modules, and the other being the particle tracking using the laminar flow and the particle tracking modules. The latter simulation was carried out to investigate the residence time distribution of the microchannel network, which is required for de-convolution of the detection signal. The fluid was assumed to be incompressible and Newtonian in the two simulations. The fluid flow governing equations [36] are shown below:(1)$$\nabla \cdot \vec{u}=0,$$
(2)$$\frac{\partial \vec{u}}{\partial t}+(\vec{u}\cdot \nabla )\vec{u}=-\frac{1}{\rho }\nabla P+\frac{\mu }{\rho }\Delta \vec{u},$$
where ρ is the fluid density, $\vec{u}$ is the fluid velocity, *μ* is the dynamic viscosity of the culture medium and *P* is the pressure. 

Due to the low lactate concentration, the transport process of lactate on the microfluidic chip, including convection and diffusion, was calculated through the dilute species transport module in COMSOL. The governing equation was as follows:(3)$$\frac{\partial c}{\partial t}-D\nabla^{2}c+\vec{u}\nabla c=0,$$
where *c* is the concentration of the lactate and *D* denotes the diffusion coefficient. The third term on the left side of Equation (3) describes the convective transport due to a velocity field $\vec{u}$.

In the particle tracking simulation, the method selected is the first-order formulation called Newtonian, ignoring inertial terms in the interface of Particle Tracing for Fluid Flow. In this formulation, the particle velocity is assigned for each time step so that the net force of each particle is zero. In this calculation, the only force on each particle is the drag force, so the particle velocity is automatically defined as $\vec{F}$_D_ = $\vec{0}$. This restores the trivial solution of the tracer particles following the fluid streamline, $\vec{u}=\vec{v}$. The model is solved in two stages. First, the fluid velocity and pressure are solved using a Stationary study step. Then the particle trajectories are computed using a Time Dependent study step. 

The motion of small particles in the fluid is governed by Newton’s second law,
(4)$$\frac{d\vec{q}}{dt}=\vec{v}\;\mathrm{and}$$
(5)$$\frac{d}{dt}(m_{p}\vec{v})={\vec{F}}_{t},$$
where $\vec{q}$ is the particle position, $\vec{v}$ is the particle velocity, *m_p_* is the particle mass and $\vec{F}$*_t_* is the total force.

In this simulation, the total force is determined by the drag force *F_D_*. Because the particles are very small, and the particle velocity relative to the fluid is not too large, Stokes drag law is applicable,
(6)$${\vec{F}}_{D}=3\pi \mu d_{p}(\vec{u}-\vec{v}),$$
where $\vec{u}$ is the fluid velocity, *μ* is the fluid dynamic viscosity and *d_p_* is the particle diameter.

The inlet velocity is consistent with the experimental conditions. At the outlet, a constant atmosphere pressure was assumed. A no-slip boundary condition was applied for all wall boundaries (i.e., the fluid has zero velocity relative to the wall). For the simulation, the Reynolds number was calculated at about 0.084, so the flow could be safely regarded as a laminar flow. For the transport of diluted species module, the initial lactate concentration of the entire fluid domain was set to 0. The inflow and outflow surfaces of the lactate were consistent with the fluid, and the inflow lactate concentration was 0. The real-time generation of lactate was replaced by the flux boundary condition in the transport of diluted species module of COMSOL, and the flux value was 5.3 × 10^−8^ mol/(m^2^·s), which was calculated from real-time lactate generation data. The other walls were set as no-flux boundaries. The diffusion coefficient of lactate in water, 1.55 × 10^−10^ m^2^/s, was used for simulation [37]. For the particle tracking module, particles are released at the outlet of the cell culture chamber, and the initial position of particles is set according to the imported matrix. The matrix can determine the number of particles and ensure that particles are evenly distributed at the release cross-section. The convergence criteria were set as 1 × 10^−6^. The backward differentiation formula (BDF) was used as the time-dependent solver to discretize the transient term. The algebraic multigrid solver, based on a finite element method, was used. The computation domain is shown in Figure 2A and the grid meshing of the 3D cell culture chamber is shown in Figure 2B, with an enlarged view showing in Figure 2C. Tetrahedral and boundary layer mesh were employed for automatic meshing. The number of meshes was about 3.6 million.

### 2.6. Convection-Diffusion

Due to the convection and diffusion of the fluid, the lactate concentration on the chip is constantly changing as the medium flows to the detection zone. In order to calculate the lactate concentration in the cell culture chamber, hydrodynamic analysis of this process was necessary. In this paper, the lactate concentration of each segment was theoretically analyzed, according to the hydrodynamic process of the fluid on the microchip. In this experimental model, the Peclet number is 2710 (Pe=Lv/D, where L is the characteristic length, v is the flow velocity and D is the diffusion coefficient). The effect of fluid flow on lactate transfer cannot be ignored. Therefore, we should not only consider the effect of diffusion on the concentration distribution of lactate, but also consider the fluid flow. Due to the non-uniform velocity distribution on the channel cross-section, elements of fluid may need different times to pass through the channel. The distribution of these times for the stream of fluid leaving the microchannel is called the exit age distribution, *E*, or the residence time distribution (RTD) of fluid. *E* has the units of time^−l^. We can divide the fluid flow on the chip into two sections. The first section is from the cell culture chamber to the intersection of the reagent solution channel and the main channel, while the second section is from the intersection point to the detection zone. The RTDs for these two sections will be calculated and designated as *E*_1_ and *E*_2_. With knowledge of the initial lactate concentration (*C*_0_) in the culture chamber, the lactate concentration, after the medium passes through these two sections, can be calculated by using of the convolution integral equation. Specifically, *C*_1_ = *C*_0_ × *E*_1_ and *C*_2_ = *C*_1_ × *E*_2_, where *C*_1_ is the lactate concentration at the intersection point, and *C*_2_ is the lactate concentration at the detection zone. The convolution integral formula is described as follows [38]:(7)$$C_{out}(t)=\int_{0}^{t}C_{in}(t^{\prime })E(t-t^{\prime })dt^{\prime },$$
where *C_in_* is the concentration distribution of lactate before flowing through the microchannel, *E* is the residence time distribution of particles and *C_out_* is the concentration distribution of lactate after flowing through the microchannel.

Finally, the lactate concentration in the cell culture chamber can be computed through de-convolution of the detected lactate concentration signal.

## 3. Results and Discussion

### 3.1. Sensor Characterization

To get an optimal detection result, the concentration of lactate oxidase (LOX) and horseradish peroxidase (HRP) are two key indexes to consider. In this experiment, the lactate concentration was set to be 200 μM and the Amplex Red concentration was 500 μm. Figure 3A,B shows the effects of LOX and HRP concentrations on detection results. It can be found that with the increase of enzyme concentration, the signal intensity gradually increases and finally tends to stabilize. The final selected enzyme concentrations were 5 U/mL LOX and 50 U/mL HRP. The signal enhancement effect is obvious based on the trends or special points. It can even reach more than 40% by only adjusting the concentration of a single enzyme. Therefore, it can greatly improve the sensitivity of the indicator and provides an important means for highly sensitive detection of lactate production.

The standard curve of lactate concentration to voltage is shown in Figure 3C. The measurement of this curve used the same experimental conditions as the cell lactate measurement experiment. The reagent solution contains 500 μM Amplex Red, 5 U/mL LOX and 50 U/mL HRP. The concentration of lactate (Sigma Aldrich, St. Louis, MO, USA) was 1 to 1000 μM. The standard curve was measured multiple times and error analysis was performed. The standard curve was drawn after subtracting the background from the obtained signal. According to the standard curve, the detection range of the indicator was 1–500 μM, where the optimal detection range was 50–300 μM. In this section, the slope of the standard curve was the largest, so it has the highest sensitivity.

Related studies have shown that when the concentration of H_2_O_2_ is lower than the Amplex Red concentration, all the H_2_O_2_ will be completely consumed. Within this range, the fluorescence intensity increases with the increase of the lactate concentration. When the concentration of H_2_O_2_ is greater than the concentration of Amplex Red, the excess H_2_O_2_ will continue to react with the fluorescent product Resorufin to further form resazurin, which has a weaker fluorescence effect [39]. This perfectly explains why the signal starts to decline when the concentration exceeds 500 μM in the standard curve. Therefore, it is necessary to ensure that the concentration of Amplex Red is always greater than H_2_O_2_ during the experiment. The cell culture medium used in this experiment contained 500 μM glucose, ensuring that lactate production was below 500 μM. The concentration of Amplex Red used was also 500 μM, so it can be ensured that Amplex Red was in excess. In addition to being catalyzed and oxidized by H_2_O_2_, Amplex Red can also react to produce Resorufin under light conditions, and light of different wavelengths has an effect on the reaction speed. After the complete reaction, continuous light will cause the fluorescent substance to continue to react, resulting in a decrease in fluorescence intensity [40]. Therefore, static detection of lactate was no longer suitable for long-term detection. In this experiment, the method of flow injection analysis was thus selected to detect the concentration of lactate under flow conditions, and the whole process of the experiment was protected from light.

### 3.2. Cell Metabolism Detection on the Microfluidic Chip

To achieve a maximal detection signal, it is imperative to ensure that the fluid flow duration exceeds the reaction time. However, excessively long fluid flow durations can prolong the detection cycle. Therefore, it is crucial to identify a flow rate that ensures both optimal detection outcomes and a brief detection period. In the detection step, the reagent solution and the medium in the cell culture room were pushed into the channel simultaneously and reacted before being fed into the detection zone for optical detection. In this step, it is necessary to ensure that the reagent completely reacts with the lactate in the medium. For example, the lactate in the medium was converted 1:1 into Resorufin with red fluorescence. This is because the signal reaches its maximum only after the reaction is complete. An important factor affecting whether the reaction can take place completely is the flow rate. Therefore, the effect of flow rate on the detection signal was tested in this experiment (Figure 4). It can be found from the experimental results that the signal did not change significantly when the flow rate was below 1 μL/min, so a flow rate of 1 μL/min was finally selected.

In order to realize pollution-free and highly sensitive detection of cell metabolism, a backflow prevention channel and a flow/stop cycle were used in this study. The backflow prevention channel can prevent cells from being polluted by reagent and buffer solutions. For the flow/stop cycle, the stop step was used for accumulation of produced lactate and the flow step was used for lactate detection as well as flushing. Each detection cycle lasts for 2 h and 40 min, including a stopping step of 2 h (stop segment), a detection step of 20 min (flow segment), and a flushing step of 20 min (flow segment). The 2-h stopping segment allows the optical signal generated by the accumulated lactate to appear in the indicator’s optimal detection interval, and the flow segment’s duration depends mainly on the channel size and flow rate.

The detection signal of lactate produced by cells measured during a flow/stop cycle is shown in Figure 5A. The signal has eight cycles in total and the detection time was more than 20 h. The system has the ability to detect cell metabolism over a long time period. Figure 5B shows the comparison of the detection data of the flow segment (including detection step and flushing step) during the eight cycles. It is noticed that the signal intensity during the stopping step is very low, mainly due to background fluorescence. When the flow starts, the signal first increases and then decreases. This is due to the fact that, during the flow segment, the lactate accumulated in the cell culture chamber reacts with the indicator in the microchannel to produce Resorufin with red fluorescence, which results in a gradually increasing signal. The signal will be reduced when Resorufin is washed away.

The detection results were closely related to the microchip structure. Their corresponding relationship is shown in Figure 6. In the detection step, the liquid that first entered the detection zone was the liquid from section I on the chip. This section of liquid did not contain lactate, and the detection result was still background fluorescence. Then what entered the detection zone was the liquid in section II. Due to diffusion, this section already contains lactate, so the detection signal increased rapidly, and the signal reached a peak value when the liquid in the culture chamber entered the detection zone. Finally, after the liquid from section III flowed to the detection zone, the signal dropped and reached a stable value. This stable value was caused by the real-time consumption of glucose to produce lactate when the section III liquid passed through the cells. In the flushing step, as the liquid of section IV flushed into the detection zone, the signal gradually becomes a background fluorescence. The formation of *P*_1_ and *P*_2_ peaks in the detection signal was caused by the residual pressure of the syringe pump and the pressure of the pipe deformation. Under the function of pressure and diffusion, the lactate in the middle channel moved to the reagent solution channel and reacted with the indicator to generate red fluorescence. Therefore, when this section of liquid flowed to the detection zone, a small peak is formed. The shape and trend of each cycle were consistent with each other. The peak value of the signal was around 2 V, and the peak value decreased, mainly due to the apoptosis of the cells on the chip. The peak value was in the optimal detection range of the indicator, and the detection signal was much larger than the background signal. It can be concluded that the chip structure, flow rate and indicator concentration in this paper were reasonable.

### 3.3. Influence of Convection-Diffusion Process on Lactate Detection

In this study, the convection-diffusion process of lactate on the microchip was simulated, and the simulation method or model is described in Section 2.5. Figure 7 shows the lactate concentration distribution in the microfluidic chip at 7200 s, 7350 s and 7550 s from the simulation results. It can be seen that there are significant changes in the lactate concentration distribution during the fluid flow process. Lactate initially accumulates in the cell culture chamber and then disperses after the flow begins. The maximum concentration decreases while the distribution range becomes wider. This means that the lactate concentration distribution detected downstream is already different from the concentration distribution in the cell culture chamber. The fluctuation of the t = 7550 s curve at around 95 mm is due to the intersection of the reagent channel and the main channel at this point, resulting in a decrease in lactate concentration.

In order to show the difference in lactate concentration between the cell culture chamber and the detection zone more clearly, Figure 8 compares the changes in lactate concentration in two regions in one detection cycle. During 0–7200 s, lactate accumulated in the culture chamber, and the concentration of lactate in the culture chamber increased. After 7200 s, the accumulated lactate was flushed out and rushed to the detection zone, so the concentration of lactate in the culture chamber decreased, while the concentration in the detection zone began to increase. Figure 8 demonstrates the contrast in lactate concentration between the detection zone and the culture chamber, where the red line represents the lactate concentration curve in the detection zone as a function of time, which is notably distinct from the lactate concentration in the culture chamber (black line). This difference primarily arises from the convection-diffusion process that occurs during the transport of the culture medium within the microchip. The substantial differences in the size and distribution of these two regions render it challenging to directly compute the lactate metabolism of the cells based on the concentrations detected in the detection zone.

In order to derive the real-time lactate metabolic rate of the cells based on the detected lactate concentration signal, we must analyze the changes in lactate concentration during this transportation process, and then deduce the lactate concentration in the cell culture chamber based on the analysis results, thereby obtaining the lactate metabolic rate. Particle tracking simulations were utilized to analyze this process. The simulation was initially optimized with respect to parameters, and the impact of the number of released particles, number of grids and time interval on the simulation results were tested. The residence time distributions of the particles were calculated using different parameters, as shown in Appendix A. Through the analysis of simulation results, we ultimately determined the optimal time interval, number of particles and number of grids for particle release as 0.5 s, 250,000 particles and 9,640,000 grids, respectively.

Utilizing the aforementioned parameters, particle tracking simulations were conducted to observe the variation of particle distribution during fluid flow. Figure 9A displays the particle trajectories in the microchannel at 95 s, with colors indicating their respective velocities. The particles released at the exit of the cell culture chamber (point A) were collected by the particle counter at point B (the junction of the main channel and the reagent channel). It was observed that particle velocities within the microchannel were not consistent, with the fastest particle moving at approximately 8 × 10^−4^ m/s, and the slowest particle moving at only 1 × 10^−4^ m/s. The significant differences in particle velocities resulted in varying time requirements for particles to travel through the microchannel, which is the primary factor affecting the particle distribution. Figure 9B illustrates the particle trajectories at 102 s, with particles being released at point B and collected by the particle counter at point C (the entrance of the detection zone). It is evident from the figure that the distribution of particle residence times varies among different particles. By means of the simulation, the particle residence time distributions from segments A to B and B to C were calculated. Based on the particle residence time distribution, the lactate concentration in the detection zone can be calculated from the lactate concentration in the cell culture zone.

### 3.4. Theoretical Analysis of the Convection-Diffusion Process

In this section, the concentration change of lactate in the medium during the fluid flow process will be analyzed. First, the lactate is generated in the cell culture chamber and undergoes a 2-h static diffusion process. The lactate is then discharged from the culture chamber at a flow rate of 1 μL/min and mixed with the reagent solution at point B. Then the mixed liquids flow to the detection zone for detection. After detection, the flushing liquid enters the microchip to flush away the residual liquid. The change of lactate concentration in the flow process is shown in Figure 10. The process is divided into two segments: A to B and B to C (as depicted in Figure 9), where *C_in_*_1_ is obtained via static diffusion and the medium subsequently flows from A to B. Owing to the brief time period (about 7 min) and the small diffusion coefficient of lactate (around 1.55 × 10^−10^ m^2^/s), the diffusion effect is neglected in this process. Here, we exclusively consider the influence of fluid flow on the distribution of lactate concentration, which can be computed via the convolution integral of Equation (7). *C_out_*_1_ is the convolution of E_1_ and *C_in_*_1_, with *E*_1_ obtained via particle tracking simulations (as illustrated in Figure 10A). *C_in_*_2_ is defined by the mixture of *C_out_*_1_ and reagent solution at point B, and *C_out_*_2_ is the convolution of *E*_2_ with *C_in_*_2_ (as depicted in Figure 10B). *C_out_*_2_ signifies the final lactate concentration impacted by convection-diffusion.

Figure 10C shows the comparison of the calculated results and the experimental data, and the two curves have good coincidence between the starting point and the peak, which proves that the calculation method of convolutional integration is correct. The curve from the beginning to the peak in Figure 10C corresponds to segment I in Figure 11, which in turn corresponds to the process of the medium flowing from the culture chamber to the detection zone. In the second half of Figure 10C, the similarity between the calculated results and the experimental data is poor. In this section, the experimental data did not directly drop to zero, but remained constant for a period of time, because the cells in the culture chamber produce lactate in real time. As the flow begins, the medium is discharged from the culture chamber and a new medium will flow in. The concentration of lactate in the new medium is zero. However, because there are cells in the culture chamber, the new medium will also contain lactate. Therefore, the signal in the second half of Figure 10C has declined, but it has not directly dropped to zero. The convolution integral method is only applicable to analyze the period before the medium reaches the detection zone, and it is inappropriate to use this method for the later part. For segments II and III in Figure 11, we use a mixed equation (Equation (8)) and a one-dimensional convection-diffusion equation (Equation (9)), to make up for the deficiencies of the convolution integral method, respectively. Segment II in Figure 11 is the washing of the medium in the cell culture chamber by the new medium, which can be approximately regarded as the mixing of the two media. Segment III corresponds to the flushing phase in the experiment. This section can be approximated as the convection-diffusion between the flushing fluid and the residual fluid in the microchannel. In this process, convection-diffusion only occurs in the flow direction, so a one-dimensional convection-diffusion equation is applicable. So far, all the curves have been analyzed. Segment I of the calculation results that we are most interested in, is highly consistent with the experimental data, and there is also good consistency in segments II and III. The process depicted in Figure 10 is the computation of the lactate concentration in the detection zone from the concentration in the culture chamber using the convolution integration method. Consequently, we can utilize the deconvolution method to deduce the lactate concentration in the culture chamber from the detected lactate concentration signal, thereby obtaining the real-time lactate production rate of the cells. Finally, the calculated lactate production rate of a cell was 19.9 fmol/min · cell, which is consistent with the published results. Table 1 summarizes the lactate production rate and detection methods in published studies. Therefore, this microfluidic chip can guarantee both the safety of the cells and the accuracy of the results.
(8)$$c=(at+Vc_{0})/(V+Qt)\;\mathrm{and}$$
(9)$$c=\frac{1}{2}c_{0}(erf\frac{\left(L_{0}-(x-vt)\right)}{\sqrt{4Dt}}+erf\frac{\left(L_{0}+(x-vt)\right)}{\sqrt{4Dt}}),$$
where $c_{0}$ is the initial concentration, $L_{0}$ is the characteristic length, *v* is the flow velocity, *D* is the diffusion coefficient, *t* is the diffusion time, *a* is the real-time generation rate of lactate, *V* is the volume of the culture chamber and *Q* is the flow rate.

## 4. Conclusions

This paper has demonstrated the long-term culture of cells and the highly sensitive and pollution-free detection of lactate from cells on a microfluidic chip. In order to ensure that the development of cells was not affected, this paper adopted the method of separating the detection zone from the cell culture chamber. When the culture medium flows from the cell culture chamber to the detection zone, the lactate concentration has changed significantly, which is mainly attributed to the mixing and convection-diffusion process of fluid. Through the analysis of these processes, the relationship between lactate concentration in the cell culture chamber and detection zone was obtained. In addition, the change process of lactate concentration was simulated and theoretically analyzed, which verified the accuracy of the detection signal in experiments. The real-time lactate production rate of cells was then calculated according to the theoretical model and measured signals. The detection platform proposed in this paper eliminates cell contamination by reagents and buffer solutions during detection, setting it apart from previous studies and enhancing its versatility. However, the platform’s low throughput can only detect one group of cells and one metabolite (lactate) at a time. In the future, we aim to design microfluidic platforms that can achieve high throughput detection and simultaneous detection of multiple metabolites.

## Figures and Tables

**Figure 1 micromachines-14-00770-f001:**
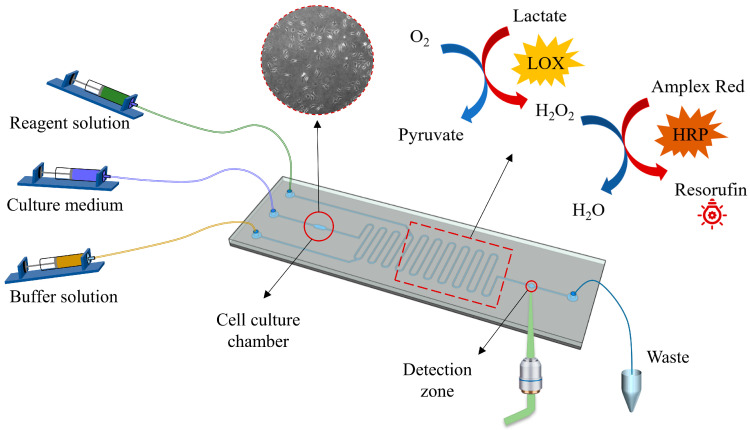
A microfluidic cell culture and detection system. The system includes a fluid driving module, a photographing module, an optical detection module and a microfluidic chip. The photographing module was placed at the bottom of the cell culture chamber, and the optical detection module was placed at the bottom of the detection zone. Moreover, there is a backflow prevention channel between the culture chamber and the detection zone (red rectangle). The size of the microfluidic chip is about 75 mm long and 25 mm wide.

**Figure 2 micromachines-14-00770-f002:**
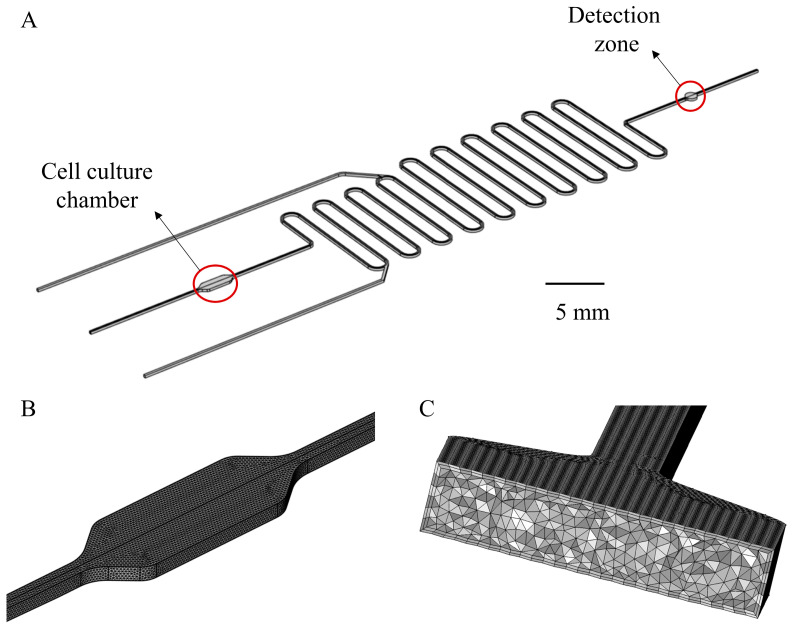
(**A**) Geometry of the Simulation model. The channel size is 200 μm × 200 μm (height × width). The width of the cell culture chamber is 1 mm and the length is 3 mm. The diameter of the detection zone is 1 mm. (**B**) Grid division detail of cell culture chamber and (**C**) enlarged cross-sectional view of the grid. Tetrahedral and boundary layer mesh were employed for automatic meshing.

**Figure 3 micromachines-14-00770-f003:**
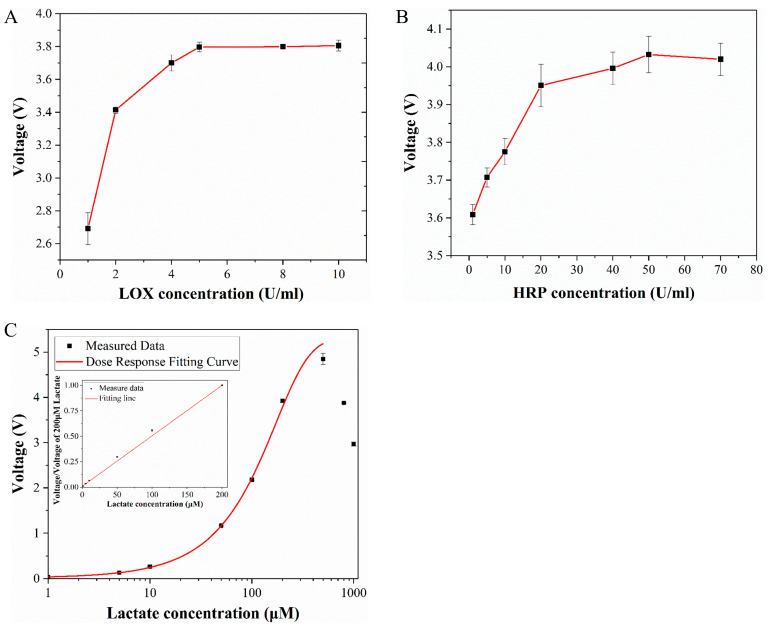
(**A**) Effect of lactate oxidase concentration on detection results. (**B**) Effect of horseradish peroxidase concentration on detection results. (**C**) Standard curve of lactate.

**Figure 4 micromachines-14-00770-f004:**
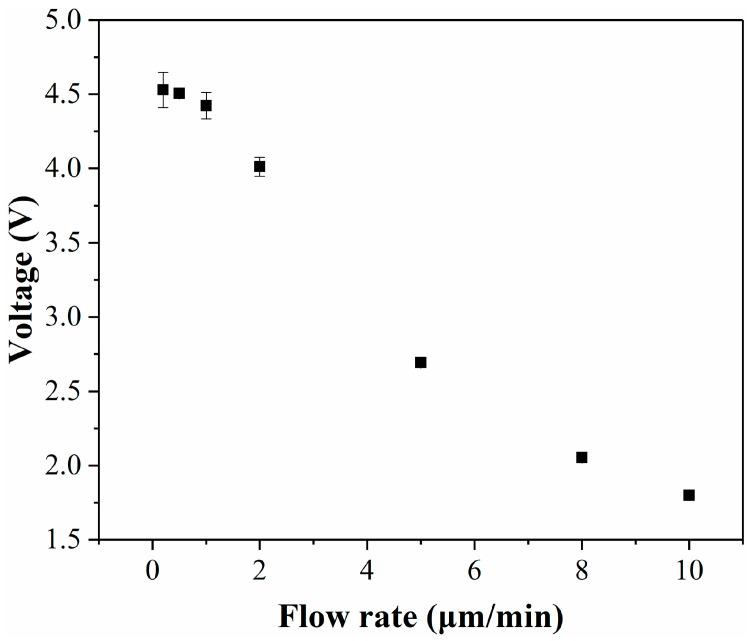
Effect of flow rate on detection results.

**Figure 5 micromachines-14-00770-f005:**
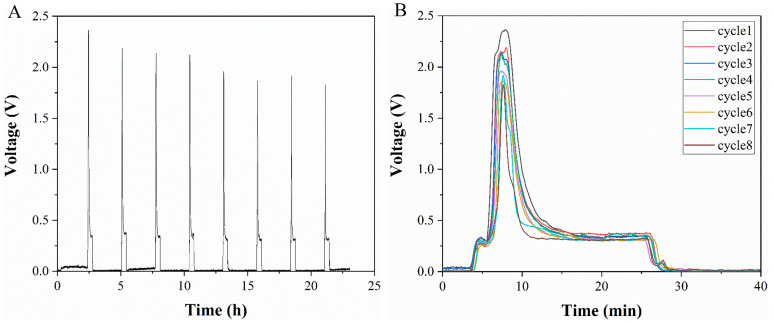
(**A**) Data of cell-generated lactate detected by flow/stop cycles. (**B**) Comparison of detection results in the flow segment of each cycle. Inset: the difference between the detection signal peaks of different cycles.

**Figure 6 micromachines-14-00770-f006:**
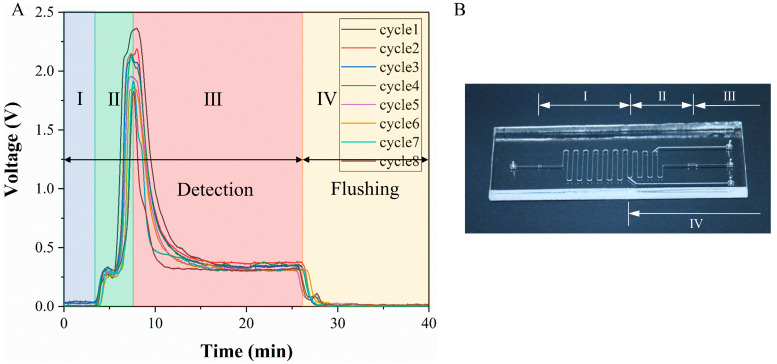
Correspondence between detection signal and chip structure. (**A**) Detection signal. (**B**) Image of the microfluidic chip.

**Figure 7 micromachines-14-00770-f007:**
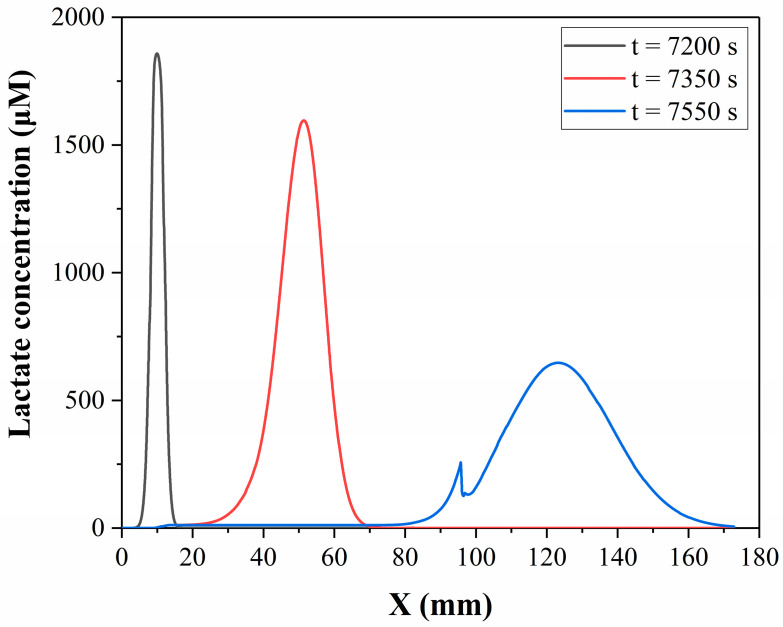
Simulated lactate concentration distribution along the centreline of the microchannel of the microfluidic chip at different times.

**Figure 8 micromachines-14-00770-f008:**
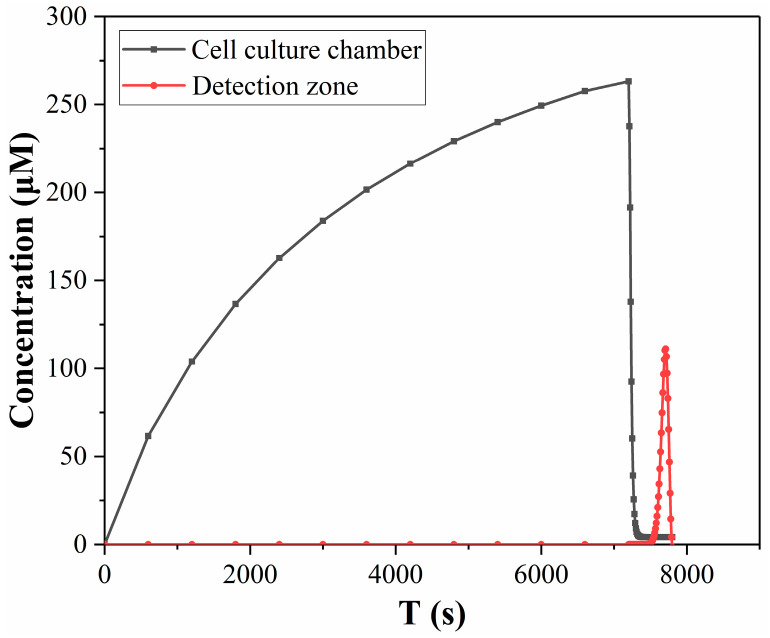
Comparison of lactate concentration in the cell culture chamber and detection zone in simulation results.

**Figure 9 micromachines-14-00770-f009:**
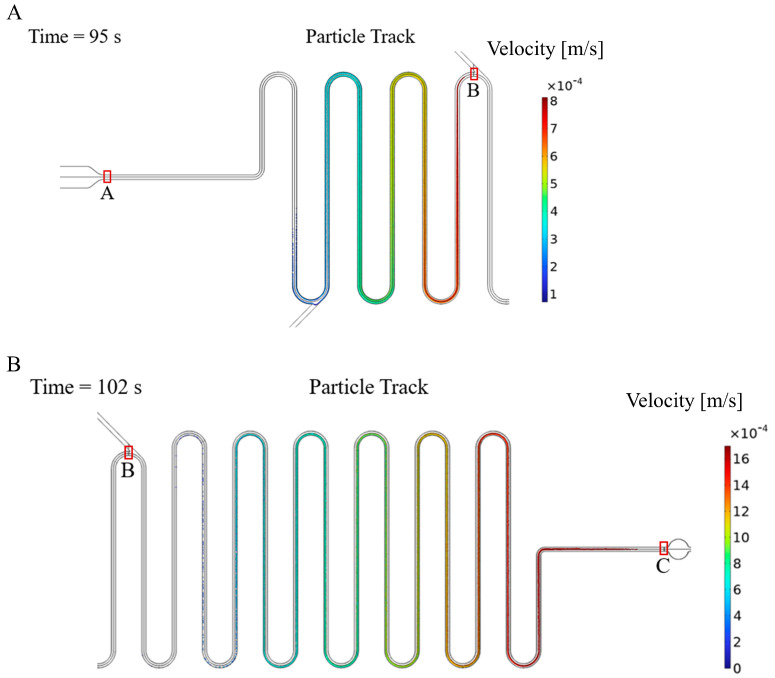
Particle trajectory when particles are released at point A (**A**) and point B (**B**), where point A is the exit of the cell culture chamber, point B is the junction of the main channel and the reagent channel and point C is the entrance of the detection zone.

**Figure 10 micromachines-14-00770-f010:**
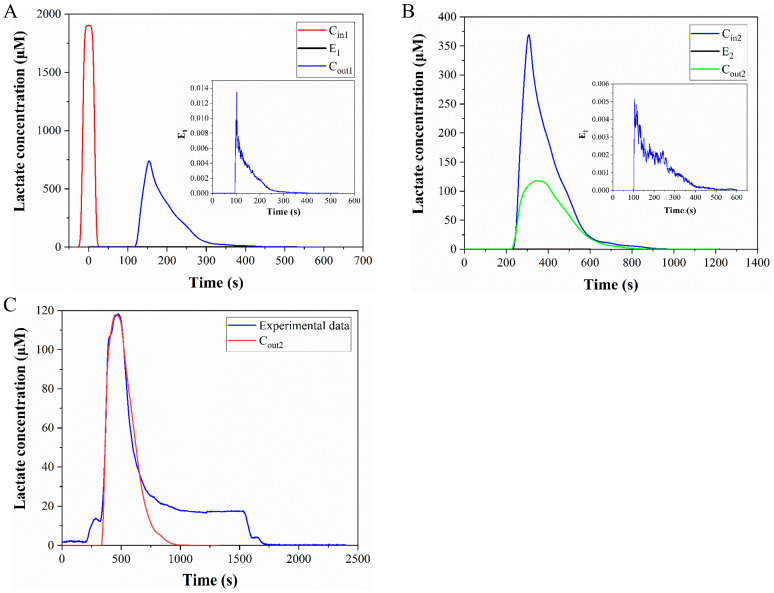
Change of lactate concentration in the flow process. (**A**) Change of lactate concentration during flow from point A to point B. (**B**) Change of lactate concentration during flow from point B to point C. (**C**) Comparison of the calculated results and the experimental data.

**Figure 11 micromachines-14-00770-f011:**
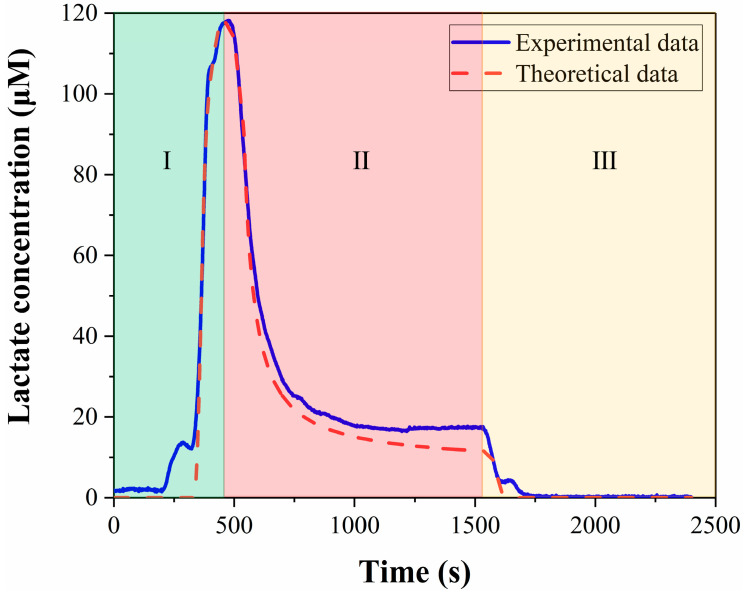
Comparison of theoretical analysis data with experimental data.

**Table 1 micromachines-14-00770-t001:** Summarized other methodologies for the lactate production measurements.

Species	Methods	Production Rates (fmol/min·Cell)	Reference
Young HUVEC	—	20	Unterluggauer [30]
Senescent HUVEC			
K562	Bioluminescent	8.9 ± 1.3	Mongersun [28]
U87 human glioblastoma cancer cell	Bioluminescent	20.4 ± 3.5	
Human umbilical vein endothelial cell	Fluorescence	19.9	This study

## Data Availability

Not applicable.

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
