# Peer review of "Pollution-Free and Highly Sensitive Lactate Detection in Cell Culture Based on a Microfluidic Chip"

_micromachines, 2023, doi:10.3390/mi14040770_

Round 1
Reviewer 1 Report
This manuscript presents a microfluidic chip composed of a backflow prevention channel, which is used for cell culture and lactate detection. This chip has good stability in metabolite quick monitoring. However, this manuscript requires revisions majors based on the following topics:
1.-The introduction should include the main limitations of the microfluidic devices for cell culture and lactate detection.
2.-What are the main advantages of the proposed microfluidic chip?
3.- The description of the 3D model should incorporate the main assumptions, boundary and load conditions, mesh, and analysis type.
4.-The resolution of Figures 3 (A-C), 4, 5, 6, 7, 8, 9, 10, and 11 must be significantly improved.
5.- The discussions of the main results shown in Figures 6, 8, 10, and 11 should be enhanced.
6.-What are the main limitations of the proposed microfluidic chip?
7.- What is the future research work?
Reviewer 2 Report
please see attached

Round 2
Reviewer 1 Report
The authors improved their manuscript based on the reviewer's comments.